# Towards Information-Seeking Agents

**Philip Bachman**[*]
phil.bachman

**Alessandro Sordoni**[*]
alessandro.sordoni

**Adam Trischler**
adam.trischler

@maluuba.com
Maluuba Research
Montréal, QC, Canada

## Abstract

We develop a general problem setting for training and testing the ability of agents to gather information efficiently. Specifically, we present a collection of tasks in which success requires searching through a partially-observed environment, for fragments of information which can be pieced together to accomplish various goals. We combine deep architectures with techniques from reinforcement learning to develop agents that solve our tasks. We shape the behavior of these agents by combining extrinsic and intrinsic rewards. We empirically demonstrate that these agents learn to search actively and intelligently for new information to reduce their uncertainty, and to exploit information they have already acquired.

## 1 Introduction

Humans possess an innate desire to know and to understand. We seek information actively through behaviors both simple (glancing at a billboard) and elaborate (conducting scientific experiments) (Gottlieb et al., 2013). These qualities equip us to deal with a complex, ever-changing environment. Artificial agents could benefit greatly from similar capacities. Discussion of information seeking behavior in artificial agents dates back at least 25 years (Schmidhuber, 1991). We contribute to this discussion by reformulating and implementing some of the associated ideas, aided by 10-20 years worth of algorithmic and computational advances. To that end, we present a general problem setting for examining the ability of models to seek information efficiently, and show that our models can apply generic information seeking behavior to improve performance in goal-oriented tasks.

Consider the game *20 Questions*. The objective is to guess the identity of some arbitrary item by asking no more than twenty yes-or-no questions (that is, collecting no more than 20 bits of information). At each turn, the questioner seeks to split the set of all viable items along some dimension, thereby shrinking the set. The "optimal" question changes from turn to turn, depending heavily on the questions asked previously. The rules of this game demand efficient seeking. Almost all "guessing games", from *20 Questions* to *Battleship* to *Hangman*, seem expressly designed to train efficient information seeking, or at least to exploit our intrinsic joy in exercising this skill.

With this in mind, we develop a collection of tasks that can be solved only through efficient acquisition of information. Our tasks vary in difficulty and complexity, but all involve searching an environment iteratively for salient fragments of information (clues) towards the fulfilment of some goal. To necessitate efficient search we impose restrictions on the information that can be acquired at each turn and on the total number of turns that an agent can take. Agents must synthesize separately acquired clues into a more complete representation of their environment in order to succeed. Our tasks are built upon several existing datasets, including *cluttered MNIST* (Mnih et al., 2014) and *CelebA* (Liu et al., 2015), as well as a *Blockworld* dataset of our own design. Through these tasks, using techniques from deep learning and reinforcement learning, we demonstrate how neural agents can be trained to seek information efficiently.

We make several contributions in this paper. First, we promote a subtle but meaningful shift in perspective regarding attention. Models that choose "where to look" (Mnih et al., 2014; Ranzato, 2014) are implicitly asking the world "what's over there?", perhaps at the loss of information received by directing the attention to another location. In contrast to this purely observational form of

questioning, we advocate a perspective that supports acquiring information from the world more actively. Rather than asking simply "what's happening?", an information-seeking model should be able to ask questions such as "what would happen if...?". Second, we develop agents that learn to exploit the information they have acquired and to look for more information when they are uncertain about the environment. Third, we show that simple task-agnostic heuristics related to the notion of information gain can be used to improve task-specific performance.

The rest of this paper is organized as follows. In Section 2 we further discuss our motivation and the relations between our problem setting and prior work. In Section 3 we formally describe the problem and the models we have devised to realize information-seeking behavior. Section 4 details our experimental results, with analysis. We conclude in Section 5.

## 2 RELATED WORK

Information seeking has been studied from a variety of perspectives, including behavioral science, psychology, neuroscience, and machine learning. In neuroscience, for instance, information-seeking strategies are often explained by biases toward novel, surprising, or uncertain events (Ranganath & Rainer, 2003). Information seeking is also a key component in formal notions of fun, creativity, and intrinsic motivation (Schmidhuber, 2010). Information seeking is closely related to the concept of attention. Both mechanisms are uniquely acts of intelligent agents, in that they do not affect the external world *per se*; rather, they alter an agent's epistemic state (Gottlieb et al., 2013). Rensink (2000) points out that humans focus attention selectively to acquire information when and where it is needed, and combine attentive fixations to build an internal representation of the environment. Similarly, attention can improve efficiency by ignoring irrelevant features outside of attended regions (Mnih et al., 2014). In this sense, attention can be considered a strategy for information seeking.

Our work thus overlaps with, and draws from, work on neural attention models – a subject which has become prominent in recent years (Larochelle & Hinton, 2010; Bahdanau et al., 2015; Ranzato, 2014; Gregor et al., 2015; Mnih et al., 2014; Sordoni et al., 2016). Larochelle & Hinton (2010), Gregor et al. (2015), and Mnih et al. (2014), for example, develop neural models that "learn where to look" to improve their understanding of visual scenes. Our work relates most closely to Mnih et al. (2014) and Gregor et al. (2015). In the RAM model (Mnih et al., 2014), visual attention is investigated through the problem of maneuvering a small sensor around a larger image in order to perform digit classification in noisy settings. DRAW (Gregor et al., 2015) uses visual attention to improve the performance of a generative model. In our work, we put tighter constraints on the amount of information that can be gathered from the environment and consider more closely whether this restricted capacity is used efficiently. We show that our model can achieve improved classification performance while operating on a significantly tighter information budget than either RAM or DRAW.[1]

Empirically, we find that a model's task-specific performance can be improved by adding a task-agnostic objective which encourages it to 1. formulate hypotheses about the state of the environment and 2. ask questions which effectively test the most uncertain hypotheses. This objective, stated more formally in Sec. 3.2, encourages the model to select questions whose answers most significantly reduce the error in the model's predictions about the answers to other questions that it might ask. In effect, this objective trains the model to simultaneously classify and reconstruct an image.

In a sense, training our models with this objective encourages them to maximize the rate at which they gather information about the environment. There exists a vast literature on applications of information gain measures for artificial curiosity, intrinsically-motivated exploration, and other more precise goals (Storck et al., 1995; Schmidhuber, 2005; Still & Precup, 2012; Hernández-Lobato et al., 2014; Mohamed & Rezende, 2015; Houthooft et al., 2016). One contribution of the current paper is to revisit some of these ideas in light of more powerful algorithmic and computational tools. Additionally, we use these ideas as a means of bootstrapping and boosting task-specific performance. I.e., we treat information seeking and curiosity-driven behavior as a developing ground for fundamental skills that a model can then apply to goal-oriented tasks.

---

[1]RAM and DRAW were not optimized for information efficiency, which biases this comparison in our favor. We're unaware of other existing models against which we can compare the performance of our models on the sorts of tasks we consider.

Current attention models assume that the environment is fully observable and focus on learning to ignore irrelevant information. Conceptually, the information-seeking approach reverses this assumption: the agent exists in a state of incomplete knowledge and must gather information that can only be observed through a restricted set of interactions with the world.

## 3 PROBLEM DEFINITION AND MODEL DESCRIPTION

We address the information seeking problem by developing a family of models which ask sequences of simple questions and combine the resulting answers in order to minimize the amount of information consumed while solving various tasks. The ability to actively integrate received knowledge into some sort of memory is potentially extremely useful, but we do not focus on that ability in this paper. Presently, we focus strictly on whether a model can effectively reason about observed information in a way that reduces the number of questions asked while solving a task. We assume that a model records all previously asked questions and their corresponding answers, perhaps in a memory whose structure is well-suited to the task-at-hand.

### 3.1 AN OBJECTIVE FOR INFORMATION-SEEKING AGENTS

We formulate our objective as a sequential decision making problem. At each decision step, the model considers the information it has received up until the current step, and then selects a particular question from among a set of questions which it can ask of the environment. Concurrently, the model formulates and refines a prediction about some unknown aspect(s) of the environment. E.g., while sequentially selecting pixels to observe in an image, the model attempts to predict whether or not the person in the image is wearing a hat.

For the current paper, we make two main simplifying assumptions: 1. the answer to a given question will not change over time, and 2. we can precisely remember previous questions and answers. Assumption 1. holds (at least approximately) in many useful settings and 2. is easily achieved with modern computers. Together, these assumptions allow us to further simplify the problem by eliminating previously-asked questions from consideration at subsequent time steps.

While the precise objectives we consider vary from task-to-task, they all follow the same pattern:

$$\underset{\theta}{\text{maximize}} \ \underset{(x,y)\sim\mathcal{D}}{\mathbb{E}} \left[ \underset{\{(q_1,a_1),...,(q_T,a_T)\}\sim(\pi_\theta,O,x)}{\mathbb{E}} \left[ \sum_{t=1}^{T} R_t(f_\theta(q_1,a_1,...,q_t,a_t),x,y) \right] \right]. \quad (1)$$

In Eqn. 1, $\theta$ indicates the model parameters and $(x,y)$ denotes an *observable/unobservable* data pair sampled from some distribution $\mathcal{D}$. We assume questions *can* be asked about $x$ and can *not* be asked about $y$. $\{(q_1,a_1),...,(q_T,a_T)\}$ indicates a sequence of question/answer pairs generated by allowing the policy $\pi_\theta$ to ask $T$ questions $q_t$ about $x$, with the answers $a_t$ provided by an observation function $O(x,a_t)$[2]. $R_t(f_\theta(\cdot),x,y)$ indicates a (possibly) non-stationary task-specific reward function which we assume to be a deterministic function of the model's *belief state* $f_\theta(\cdot)$ at time $t$, and the observable/unobservable data $x/y$. For our tasks, $R_t$ is differentiable with respect to $f_\theta$, but this is not required in general. Intuitively, Eqn. 1 says the agent should ask questions about $x$ which most quickly allow it to make good predictions about $x$ and/or $y$, as measured by $R_t$.

As a concrete example, $(x,y)$ could be an image/annotation pair sampled from $\mathcal{D} \equiv$ CelebA, each question $q_t$ could indicate a 4x4 block of pixels in $x$, $O(x,q_t)$ could provide the value of those pixels, and $R_t$ could be the log-likelihood which the model's belief state $f_\theta$ assigns to the true value of $y$ after observing the pixels requested by questions $\{q_1,...,q_t\}$ (i.e. $\{a_1,...,a_t\}$).

### 3.2 TRAINING

We train our models using Generalized Advantage Estimation (Schulman et al. (2016), abbr. GAE), TD($\lambda$) (Sutton & Barto (1998)), and standard backpropagation. We use GAE to train our models how to *make better decisions*, TD($\lambda$) to train the value function approximators required by GAE, and backpropagation to train our models how to *cope with their decisions*. When the observation

---

[2]I.e., we may be unable to backprop through $O(x,a)$, though its derivatives could be useful when available.

function $O(x, q_t)$ is differentiable w.r.t. $q_t$ and the policy $\pi_\theta(q_t|h_{:t})$ has a suitable form, GAE and TD($\lambda$) can be replaced with a significantly lower variance estimator based on the "reparametrization trick" (Kingma & Welling, 2014; Rezende et al., 2014; Silver et al., 2014).

We train our models by stochastically ascending an approximation of the gradient of Eqn. 1. Considering a fixed $(x, y)$ pair – incorporating the expectation over $(x, y) \sim \mathcal{D}$ is simple – the gradient of Eqn. 1 w.r.t. $\theta$ can be written:

$$\nabla_\theta \underset{\{(q_t, a_t)\} \sim (\pi_\theta, O, x)}{\mathbb{E}} \left[ \sum_{t=1}^{T} R_t(f_\theta(q_1, a_1, ..., q_t, a_t), x, y) \right] =$$

$$\underset{\{(q_t, a_t)\} \sim (\pi_\theta, O, x)}{\mathbb{E}} \left[ \sum_{t=1}^{T} (\nabla_\theta \log \pi_\theta(q_t|h_{:t}) R_{t:} + \nabla_\theta R_t(f_\theta(h_{:t+1}), x, y)) \right], \qquad (2)$$

in which $R_{t:}$ refers to the total reward received after asking question $q_t$, and $h_{:t}$ indicates the history of question/answer pairs $\{(q_1, a_1), ..., (q_{t-1}, a_{t-1})\}$ observed prior to asking question $q_t$.

The gradient in Eqn. 2 can be interpreted as comprising two parts:

$$\nabla_\theta^\pi \equiv \underset{\{(q_t, a_t)\} \sim (\pi_\theta, O, x)}{\mathbb{E}} \left[ \sum_{t=1}^{T} \nabla_\theta \log \pi_\theta(q_t|h_{:t})(R_{t:} - V_\theta(h_{:t})) \right] \quad \text{and} \qquad (3)$$

$$\nabla_\theta^f \equiv \underset{\{(q_t, a_t)\} \sim (\pi_\theta, O, x)}{\mathbb{E}} \left[ \sum_{t=1}^{T} \nabla_\theta R_t(f_\theta(h_{:t+1}), x, y) \right], \qquad (4)$$

where we have introduced the approximate value function (i.e. baseline) $V_\theta(h_{:t})$. Roughly speaking, $V_\theta(h_{:t})$ provides an estimate of the expectation of $R_{t:}$ and is helpful in reducing variance of the gradient estimator in Eqn. 3 (Sutton & Barto, 1998). Intuitively, $\nabla_\theta^\pi$ modifies the distribution of question/answer sequences $\{(q_1, a_1), ..., (q_T, a_T)\}$ experienced by the model, and $\nabla_\theta^f$ makes the model better at predicting given the current distribution of experience. Respectively, $\nabla_\theta^\pi$ and $\nabla_\theta^f$ train the model to make better decisions and to cope with the decisions it makes.

We estimate $\nabla_\theta^f$ directly using standard backpropagation and Monte Carlo integration of the required expectation. In contrast, obtaining useful estimates of $\nabla_\theta^\pi$ is quite challenging and a subject of ongoing research. We use the GAE estimator presented by Schulman et al. (2016), which takes a weighted average of all possible $k$-step actor-critic estimates of $R_{t:}$. Details are available in the supplementary material.

### 3.3 EXTRINSIC AND INTRINSIC REWARD

The specification of the reward function $R_t$ is a central aspect of sequential decision making problems. Extrinsic rewards incorporate any external feedback useful to solve the problem at hand. For example, $R_t$ may reflect the log-likelihood the model assigns to the true value of the unknown target $y$, as in Mnih et al. (2014), or some performance score obtained from the external environment, as in Silver et al. (2014). Because extrinsic rewards may be sparse, intrinsically motivated reinforcement learning (Chentanez et al., 2004; Mohamed & Rezende, 2015) aims to provide agents with reward signals that are task-agnostic and motivated rather by internal drives like curiosity.

In our work, we use a reward function $R_t(.) = r_t^E(.) + r_t^I(.)$, which sums an extrinsic and intrinsic reward function respectively. We suppose that the belief state $f_\theta$ comprises a probabilistic model $q(x|f_\theta(.))$ of the unobserved world $x \sim \mathcal{D}$. Therefore, we use an intrinsic reward given by the negative cross-entropy $r_t^I = \mathbb{E}_{x \sim \mathcal{D}}[\log q(x|f_\theta(h_{:t}))]$, which encourages the model to form an accurate belief about the world distribution $\mathcal{D}$.

Instead of using the same intrinsic reward for each question that has been asked, we reward each question that the model asks by the difference in the rewards $r_{t+1}^I - r_t^I$, which is the difference in cross-entropy between the model beliefs after the question has been asked and those prior to the question $q_t$. Intuitively, this intrinsic reward has the effect of more explicitly favoring the questions whose answers provide the most useful information about the underlying world $x$.

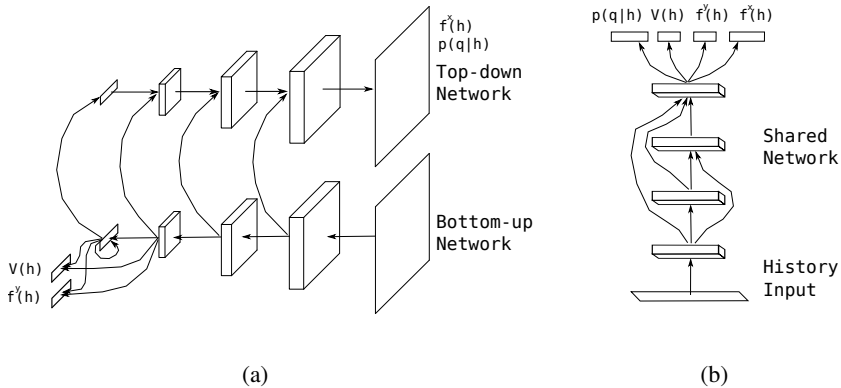

(a) (b)

Figure 1: The network architectures developed for (a) our experiments with images and (b) our experiments with permutation-invariant data. We describe how computation proceeds through these architectures at each step of a trial in Section 3.4. The current trial history $h_{:t}$ is input to (a) through the bottom-up network. The values $f_\theta^x(h_{:t})$, $f_\theta^y(h_{:t})$, $V_\theta(h_{:t})$, and $\pi_\theta(q_t|h_{:t})$ are collected from the indicated locations. We use $f_\theta^{x,y}$ to denote a model's predictions about the complete observable data $x$ and the unobservable data $y$. Computation in architecture (b) proceeds from bottom to top, starting with the history input $h_{:t}$ and then passing through several fully-connected layers linked by shortcut connections. The values $f_\theta^x(h_{:t})$, $f_\theta^y(h_{:t})$, $V_\theta(h_{:t})$, and $\pi_\theta(q_t|h_{:t})$ are computed as linear functions of the output of the shared network.

### 3.4 MODEL ARCHITECTURES FOR INFORMATION SEEKING

We use deep neural networks to represent the functions $\pi_\theta$, $f_\theta$, and $V_\theta$ described in the preceding sections. Our networks share parameters extensively. Figure 1 illustrates the specific architectures we developed for tasks involving images and generic data. For the tasks we examine, the number of possible questions is moderately sized, i.e. $< 1000$, and the possible answers to each question can be represented by small vectors, or perhaps even single scalars. Additionally, the response to a particular question will always have the same type, i.e. if question $q_t$ produces a 4d answer vector $a_t \equiv O(x, q_t)$ for some $x$, then it produces a 4d answer vector for all $x$.

Given these assumptions, we can train neural networks whose inputs are tables of (qrepr, answer) tuples, where each tuple provides the answer to a question (if it has been asked), and a *question representation*, which provides information about the question which was asked. E.g., a simple question representation might be a one-hot vector indicating which question, among a fixed set of questions, was asked. Our networks process the tables summarizing questions and answers which they have observed so far (i.e. a trial history $h_{:t}$) by vectorizing them and feeding them through one or more hidden layers. We compute quantities used in training from one or more of these hidden layers – i.e. the value function estimate $V_\theta(h_{:t})$, the policy $\pi_\theta(q_t|h_{:t})$, and the belief state $f_\theta(h_{:t})$.

The architecture in Fig. 1a first evaluates a *bottom-up* network comprising a sequence of convolutional layers topped by a fully-connected layer. Each convolutional layer in this network performs 2x downsampling via strided convolution. For the fully-connected layer we use an LSTM (Hochreiter & Schmidhuber, 1997), which maintains internal state from step to step during a trial. After computing the output of each layer in the bottom-up network, we then evaluate a sequence of layers making up the *top-down* network. Each layer in the top-down network receives input both from the preceding layer in the top-down network and a partner layer in the bottom-up network (see arrows in (a)). Each convolutional layer in the top-down network performs 2x upsampling via strided convolution.

The input to the bottom-up network is an image, masked to reveal only the pixels whose value the model has previously queried, and a bit mask indicating which pixels are visible. The output of the top-down network has the same spatial shape as the input image, and provides both a reconstruction of the *complete* input image and the values used to compute $\pi_\theta(q_t|h_{:t})$. The value function $V_\theta(h_{:t})$ at time $t$ is computed as a linear function of the bottom-up network's LSTM state. For labelling tasks, the class prediction $f_\theta^y(h_{:t})$ is computed similarly. The input reconstruction $f_\theta^x(h_{:t})$ is taken from the

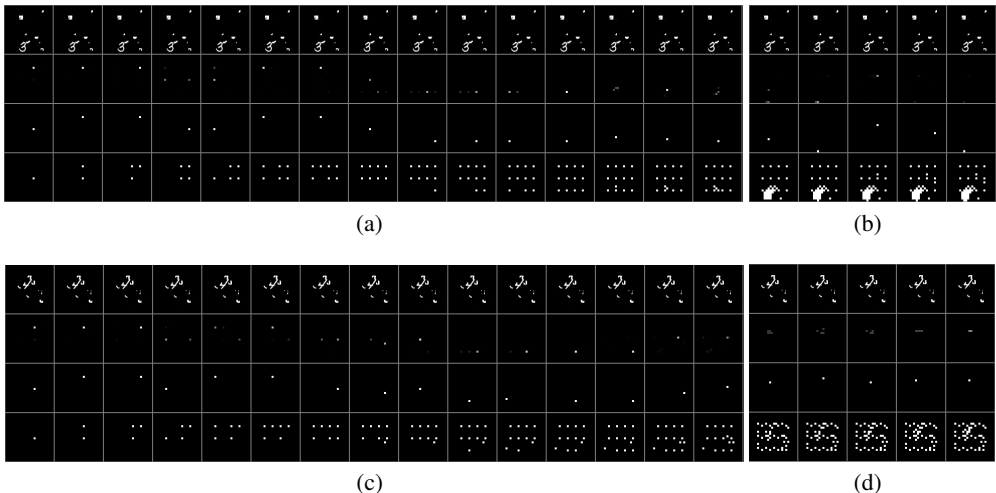

Figure 2: Model behavior on cluttered MNIST task. Zoom in for best viewing. (a) and (c) show the first 15 peeks made by a model in a success and failure case, respectively. (b) and (c) show the corresponding final 5 peeks for each of these trials. The model has learned a fairly efficient searching behavior, and is generally able to locate the digit and then concentrate its resources on that location. Occasionally the presence of clutter disrupts the search process and the model does not have enough resources to recover. When the model is given access to global summary info, it concentrates its resources immediately on the correct location with little or no search.

top-down network's output. The policy $\pi_\theta(h_{:t})$ is computed by summing pixel-level values from a single channel in the top-down network's output over regions whose pixels the model has yet to ask about. A softmax is applied to the summed pixel-level values to get the final probabilities for which pixels the model will ask about next.

The fully-connected architecture in Fig. 1b performs similar computations to the convolutional architecture. The input to this model is an observation vector, masked to reveal only the features which have already been requested by the model, and a mask vector indicating which features have been requested. These vectors are concatenated and fed through a sequence of four shared hidden layers linked by shortcut connections. The values $f_\theta^x(h_{:t})$, $f_\theta^y(h_{:t})$, $V_\theta(h_{:t})$, and $\pi_\theta(q_t|h_{:t})$ are all computed as linear functions of the output of the final shared layer, with a softmax used to make $\pi_\theta(q_t|h_{:t})$ a distribution over which feature to request next.

We use Leaky ReLU activation functions throughout our networks, and apply layer normalization in all hidden layers to deal with the variable input magnitude caused by large variation in the number of visible pixels/features over the course of a trial. Layer weights are all initialized with a basic orthogonal init. We use the ADAM optimizer (Kingma & Ba, 2015) in all our tests, while working with minibatches of 100 examples each.

## 4 EXPERIMENTS

### 4.1 TASK 1: CLUTTERED MNIST

The first task we examine is *cluttered* MNIST classification as proposed by Mnih et al. (2014). In this task, a model must classify an MNIST digit placed uniformly at random on a 104x104 canvas[3]. Note that an MNIST digit is 28x28, so the salient information for classifying the digit occupies at most 7.3% of the full canvas. To make the task more challenging, eight pieces of clutter are randomly distributed over the canvas. Each piece is an 8x8 patch extracted from a random location in a randomly selected MNIST digit. We did not include the intrinsic reward signal during these tests.

---

[3]The original task uses a 100x100 canvas. We wanted the canvas size to be divisible by 8.

The lowest test errors previously reported on this task were 8.11% with eight four-scale glimpses of size 12x12 for RAM, and 3.36% for DRAW with eight one-scale glimpses of size 12x12. Respectively, these results consume 4608 and 1152 scalar values worth of information from the image. Both methods can cut this information use in half with about 1% increase in error.

The first model we applied to this task was structured precisely as shown in Fig. 1a. At each step of a trial, the model predicted the class of the digit in the image and selected a 4x4 block of pixels whose values it wanted to observe. These pixels were made visible to the model at the beginning of the next step. We allowed the model to view up to 41 4x4 pixel blocks. This makes 656 pixels worth of information, which is roughly comparable to the amount consumed by DRAW with four 12x12 peeks. After 41 peeks our model had 4.5% error on held-out examples from the test set. Qualitatively, this model learned a reasonably efficient search strategy (see Fig. 2) but occasionally failed to find the true digit before time ran out.

Based on the observation that sometimes purely-local information, due simply to bad luck, can be inadequate for solving this task quickly, we provided our model with an additional source of "summary" information. This summary was fed into the model by linearly downsampling the full 104x104 canvas by 8x, and then appending the downsampled image channel-wise to the inputs for appropriately-shaped layers in both the bottom-up and top-down networks. This summary comprised 13x13=169 scalar values. Provided with this information, the task became nearly trivial for our model. It was able to succesfully integrate the summary information into its view of the world and efficiently allocate its low-level perceptual resources to solve the task-at-hand. After just 10 4x4 peeks, this model had 2.5% test error. By 20 peeks, this dropped to 1.5%. After 10 peeks the model had consumed just 329 pixels worth of information – about half that of the most efficient DRAW model.

## 4.2 TASK 2: POSITIONAL REASONING FOR BLOCKWORLD

We designed *BlockWorld* to train and test inference capabilities in information-seeking agents. This synthetic dataset consists of 64x64-pixel images that depict elementary shapes of different colors and sizes. Shapes are drawn from the set $\mathcal{S} = \{triangle, square, cross, diamond\}$ and can be colored as $\mathcal{C} = \{green, blue, yellow, red\}$. The scale of the shapes may vary, with the longest border fixed to be either 12 pixels or 16 pixels.

Distinct "worlds" – the environments for our agents – are generated by sampling three objects at random and placing them on the 64x64 image canvas such that they do not intersect. Objects in each world are uniquely identifiable – we enforce that an image does not contain two objects with the same shape and color. An agent's goal in *Blockworld* is to estimate whether a specific positional statement holds between two objects in a given image. Relation statements are structured as triples $\mathcal{F} = \{(s_1, r, s_2)\}$, where $s_1, s_2 \in \mathcal{S} \times \mathcal{C}$ are shape descriptions and $r$ is a positional relation from $\mathcal{R} = \{above, below, to\ the\ right\ of, to\ the\ left\ of\}$. The query (*yellow triangle*, *above*, *yellow sphere*) is an example. Generation of true statements is straightforward: it is sufficient to pick two objects in the image and compare their coordinates to find one relation that holds. We generate negative statements by corrupting the positive ones. Given a statement triple with positive answer, we either change the relation $r$ or one property (either colour or shape) of $s_1$ or $s_2$. Thus, a statement may be false if the relation between two shapes in the world does not hold or if one of the shapes does not appear in the world (but a similar one does).

The input to our model during training is a triple $(x, s, y)$, where $s$ is a 20-dimensional multi-hot vector encoding the particular statement (16 dimensions for color and shape for the two objects and 4 dimensions for the relation) and $y$ is a binary variable encoding its truth value in image $x$. We approach the task by estimating a conditional policy $\pi_\theta(q_t|h_{:t}, a)$. Here, each dimension of the statement vector $s$ is treated as an additional channel in the bottom-up and the top-down convolutions. In practice we condition the convolutional layers at a coarser resolution (in order to limit computational costs) as follows: we form a 10x16x16 "statement" feature map by repeating $s$ along the dimensions of the image down-sampled at 4x resolution; then we concatenate these additional channels onto the output of the 4x down-sampling (up-sampling) layer in the bottom-up (top-down) convolutional stacks.

Figure 3 illustrates 13 questions asked by the model in order to assess the truth of the statement "the blue triangle is above the yellow diamond", with respect to the world pictured in the first row (see

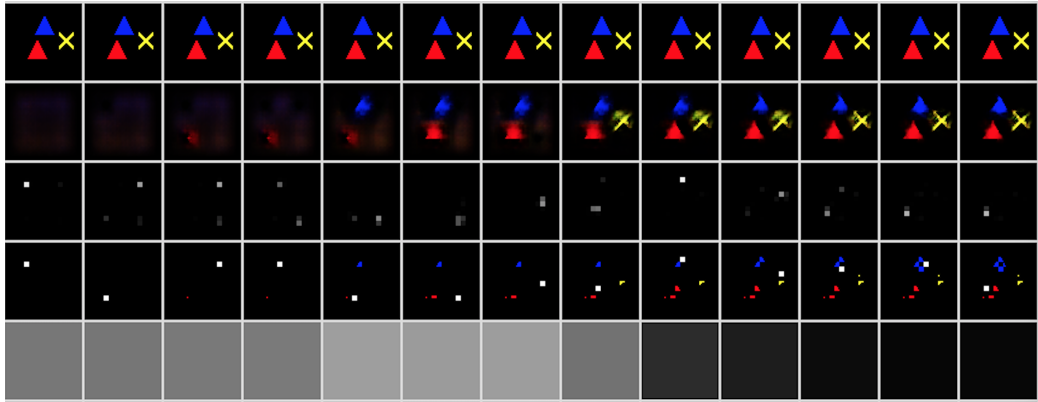

Figure 3: Impression of the model behavior for a randomly generated world and the statement "the blue triangle is above the yellow diamond". Each column is a question asked by the model. The rows correspond respectively to: 1) the input image; 2) the model reconstruction; 3) the model policy on the next possible actions; 4) the answers that the model received until that step along with the chosen question (white square); 5) the probability that the statement is true at each time-step.

figure caption for more details). At first, the model exhibits exploratory behavior, until it finds the red triangle at the 5th iteration (see the reconstructions in the second row). As it appears on the top of the frame, the model becomes confident of the truth value of the statement. At its 8th question, the yellow cross is found, which entails a drop in the confidence that the statement is true. The model finishes the prediction by tracing the objects in the image. In 10 steps, the model has consumed only 4% of the total information contained in the image (each question is worth 16 pixels). Our model with 20 questions (8% of the original image) achieves an accuracy of 96%, while a similar bottom-up convolutional architecture that takes the whole image as input (upper-bound) gets an accuracy of 98.2%. The cases in which the model makes mistakes correspond to unfruitful exploration, i.e. when the model cannot find the object of interest in the image. We also ran our model but with a random policy of question asking (lower-bound) and got an accuracy of 71%.

## 4.3    TASK 3: CONDITIONAL CLASSIFICATION FOR CELEBA

We also test our model on a real-word image dataset. *CelebA* (Liu et al., 2015) is a corpus of thousands of celebrity face images, where each image is annotated with 40 binary facial attributes (*e.g.*, "attractive", "male", "gray hair", "bald", "has beard", "wearing lipstick"). We devise a conditional classification task based on this corpus wherein the agent's goal is to determine whether a face has a given attribute. Through training, an agent should learn to adapt its questioning policy to the distinct attributes that may be queried.

In order to ensure that the learned conditional information-seeking strategy is interpretable, we exclude from the task those attributes whose presence might be ambiguous (such as 'is old' and 'pointy nose') and query only a subset of 10 attributes that can be discriminated from a specific image region. Our query attributes are "eyeglasses", "wearing hat", "narrow eyes", "wearing earrings", "mouth slightly open", "male", "wearing lipstick", "young", "wavy hair", "bald". As is common in previous works (Radford et al., 2015), we center-crop and resize images to 64 by 64. In order to condition our policy, we adopt an approach similar to the previous section.

We show the behavior of the model in Figure 4. In this case, the model is pretty effective as it reaches an accuracy of 83.1% after 2 questions (32 pixels, less than 1% of the image), 85.9% after 5 questions (90 pixels) and 87.1% after 20 questions, while the random policy scores 76.5%. The "upper-bound" architecture having access to all the image scores 88.3%.

## 4.4    TASK 4: GUESSING CHARACTERS IN HANGMAN

We also test our model in a language-based task inspired by the well-known text game *Hangman*. We sample sub-sequences of 16 characters from the Text8 corpus and train our model to guess all the

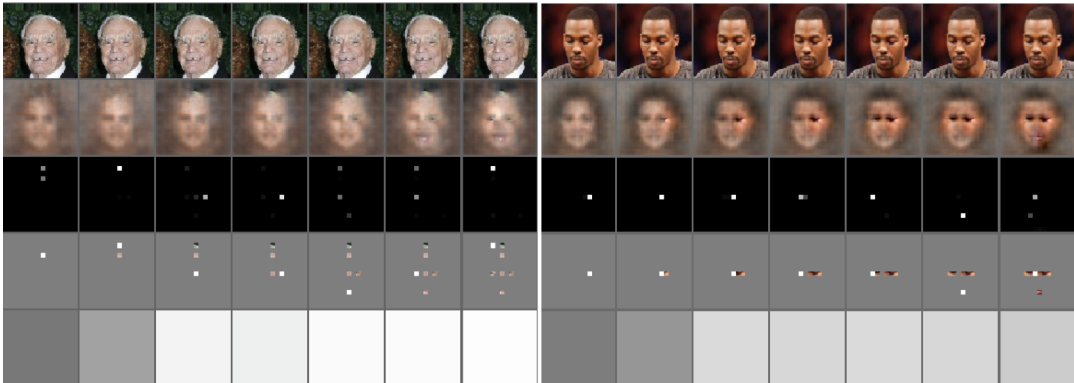

Figure 4: Conditional classification for CelebA (for a description of the rows, see Figure 3). In the left image, the model correctly guesses that the attribute "bald" is true. In right figure, the model makes a mistake about the attribute "narrow eyes", even if it correctly identifies the most relevant part to discriminate the attribute.

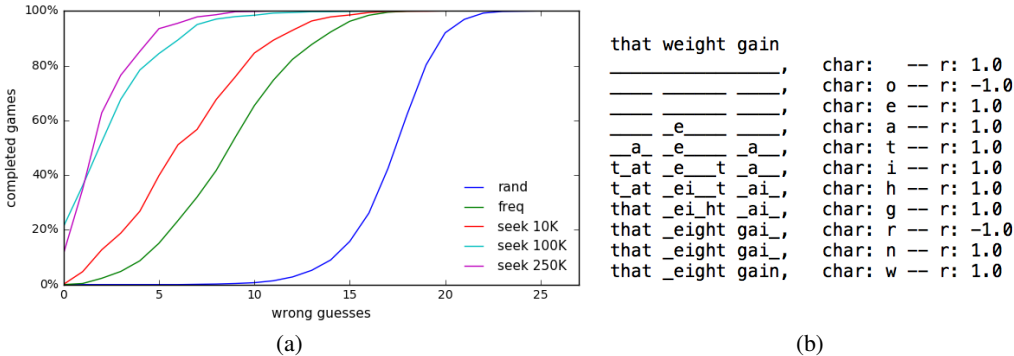

Figure 5: (a) Cumulative distribution of completed *Hangman* games with respect to the number of wrong guesses. (b) Example execution of the hangman game guessing *that weight gain* with the associated sequence of guesses and rewards.

characters in the input sequence. At each step, the model guesses a character and the environment shows all the positions in which the selected character appears. If the model asks for a character that is not in the sequence, it suffers a fixed loss of -1. On the contrary, if the character is present, the model gets a reward of +1. The main objective for the model is to maximize the expected reward.

We adapt our architecture by substituting the 2-D convolutions with 1-D convolutions, similar to Zhang et al. (2015). In Figure 5, we report the cumulative distribution of completed games with respect to the number of wrong guessed characters. *rand* is equipped with a random policy; *freq* follows the unigram distribution of characters in the training corpus; *seek* is our model after 10K, 100K and 250K updates respectively.

## 5 CONCLUSION

In this work we developed a general setting to train and test the ability of agents to seek information. We defined a class of problems in which success requires searching, in a partially-observed environment, for fragments of information that must be pieced together to achieve some goal. We then demonstrated how to combine deep architectures with techniques from reinforcement learning to build agents that solve these tasks. It was demonstrated empirically that these agents learn to search actively and intelligently for new information to reduce their uncertainty, and to exploit information they have already acquired. Information seeking is an essential capability for agents acting in complex, ever-changing environments.

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
