# Peer review of "Towards Information-Seeking Agents"

_ICLR 2017 — rejected_

[Official Review · AnonReviewer3 · rating 4 · confidence 4 · 15 Dec 2016]
**No Title**

This paper proposed to use Generalized Advantage Estimation (GAE) to optimize DNNs for information seeking tasks. The task is posed as a reinforcement learning problem and the proposed method explicitly promotes information gain to encourage exploration.

Both GAE and DNN have been used for RL before. The novelty in this paper seems to be the explicit modeling of information gain. However, there is insufficient empirical evidence to demonstrate the benefit and generality of the proposed method. An apple to apple comparison to previous RL framework that doesn't model information gain is missing. For example, the cluttered MNIST experiment tried to compare against Mnih et al. (2014) (which is a little out dated) with two settings. But in both setting the input to the two methods are different. Thus it is unclear what contributed to the performance difference.

The experiment section is cluttered and hard to read. A table that summarizes the numbers would be much better.

[Official Review · AnonReviewer1 · rating 4 · confidence 4 · 16 Dec 2016]
**Promising but unfinished paper**

This paper proposes a setting to learn models that will seek information (e.g., by asking question) in order to solve a given task. They introduce a set of tasks that were designed for that goal. They show that it is possible to train models to solve these tasks with reinforcement learning.

One key motivation for the tasks proposed in this work are the existence of games like 20Q or battleships where an agent needs to ask questions to solve a given task. It is quite surprising that the authors do not actually consider these games as potential tasks to explore (beside the Hangman). It is also not completely clear how the tasks have been selected. A significant amount of work has been dedicated in the past to understand the property of games like 20Q (e.g., Navarro et al., 2010) and how humans solve them.  It would interesting to see how the tasks proposed in this work distinguish themselves from the ones studied in the existing literature, and how humans would perform on them.  In particular, Cohen & Lake, 2016m have recently studied the 20 questions games in their paper “Searching large hypothesis spaces by asking questions” where they both evaluate the performance of humans and computer. I believe that this paper would really benefits from a similar study.

Developing the ability of models to actively seek for information to solve a task is a very interesting but challenging problem. In this paper, all of the  tasks require the agent to select a questions from a finite set of clean and informative possibilities. This allows a simpler analysis of how a given agent may perform but at the cost of a reducing the level of noise that would appear in more realistic settings.

This paper also show that by using a relatively standard mix of deep learning models and reinforcement learning, they are able to train agents that can solve these tasks in the way it was intended to. This validates their empirical setting but also may exhibit some of the limitation of their approach; using relatively toy-ish settings with perfect information and a fixed number of questions may be too simple. 

While it is interesting to see that their agent are able to perform well on all of their tasks, the absence of baselines limit the conclusions we can draw from these experiments. For example in the Hangman experiment, it seems that the frequency based model obtains promising performance. It would interesting to see how good are baselines that may use the co-occurrence of letters or the frequency of character n-grams.


Overall, this paper explores a very interesting direction of research and propose a set of promising tasks to test the capability of a model to learn from asking question. However, the current analysis of the tasks is a bit limited, and it is hard to draw any conclusion from them. It would be good if the paper would focus more on how humans perform on these tasks, on strong simple baselines and on more tasks related to natural language (since it is one of the motivation of this work) rather than on solving them with relatively sophisticated models.

[Official Review · AnonReviewer2 · rating 6 · confidence 4 · 20 Dec 2016]
**Review of "Towards Information Seeking Agents"**

Pros:

* The general idea behind the paper seems pretty novel and potentially quite cool.
* The specific technical implementation seems pretty reasonable and well-thought through.
* The general types of the tasks that they try out their approach on spans a wide and interesting spectrum of cognition abilities. 
* The writing is pretty clear.  I basically felt like I could replicate much of what they did from their paper descriptions. 


Cons:

* The evaluation of the success of these ideas, as compared to other possible approaches, or as compared to human performance on similar tasks, is extremely cursory. 

* The specific tasks that they try are quite simple.   I really don't know whether their approach is better than a bunch of simpler things on these tasks.   

Taking these two cons together, it feels like the authors basically get the implementation done and working somewhat, and then just wrote up the paper.  (I know how it feels to be under a deadline without a complete set of results.)   If the authors had used their approach to solve an obviously hard problem that previously was completely unsolved, even the type of cursory evaluation level chosen here would have been fine.  Or if they had done a very thorough evaluation of a bunch of standard models on each task (and humans too, ideally), and compared their model to those results, that would have been great.  But given the complexity of their methods and the fact that the tasks are either not well-known benchmarks or very challenging as such, it's really hard to tell how much of an advance is made here.     But it does seem like a potentially fruitful research direction.

[Final Decision · Program Chairs · 06 Feb 2017]
**ICLR committee final decision**

This paper proposes information gain as an intermediate reward signal to train deep networks to answer questions. The motivation and model are interesting, however the experiments fail to deliver. There is a lack of comparative simple baselines, the performance of the model is not sufficiently analyzed, and the actual tasks proposed are too simple to promise that the results would easily generalize to more useful tasks. This paper has a lot of good directions but definitely requires more work. I encourage the authors to follow the advice of reviewers and explore the various directions proposed so that this work can live up to its potential.